# Real-Time Gamma Radioactive Source Localization by Data Fusion of 3D-LiDAR Terrain Scan and Radiation Data from Semi-Autonomous UAV Flights

**DOI:** 10.3390/s22239198

**Published:** 2022-11-26

**Authors:** Stephan Schraml, Michael Hubner, Philip Taupe, Michael Hofstätter, Philipp Amon, Dieter Rothbacher

**Affiliations:** 1AIT Austrian Institute of Technology GmbH, Giefinggasse 4, 1210 Vienna, Austria; 2RIEGL Laser Measurement Systems GmbH, Riedenburgstr. 48, 3580 Horn, Austria; 3CBRN Protection GmbH, 1200 Vienna, Austria

**Keywords:** real-time gamma source localization, hazard detection, measurements of CBRN agents, 3D-LiDAR terrain modelling, data fusion, optimization

## Abstract

Rapid and accurate reconnaissance in the event of radiological and nuclear (RN) incidents or attacks is vital to launch an appropriate response. This need is made stronger by the increasing threat of RN attacks on soft targets and critical infrastructure in densely populated areas. In such an event, even small radioactive sources can cause major disruption to the general population. In this work, we present a real-time radiological source localization method based on an optimization problem considering a background and radiation model. Supported by extensive real-world experiments, we show that an airborne system using this method is capable for reliably locating category 3–4 radioactive sources according to IAEA safety standards in real time from altitudes up to 150 m. A sensor bundle including a LiDAR sensor, a Gamma probe as well as a communication module was mounted on a UAV that served as a carrier platform. The method was evaluated on a comprehensive set of test flights, including 28 flight scenarios over 316 min using three different radiation sources. All additional gamma sources were correctly detected, multiple sources were detected if they were sufficiently separated from each other, with the distance between the true source position and the estimated source averaging 17.1 m. We also discuss the limitations of the system in terms of detection limit and source separation.

## 1. Introduction

Rapid and accurate reconnaissance in the event of radiological and nuclear (RN) incidents or attacks is vital to launch an appropriate response. This need is made stronger by the increasing threat of RN attacks on soft targets and critical infrastructure in densely populated areas. Even small radioactive sources can cause major disruption to the general population. This work focuses on unmanned airborne localization of point sources emitting gamma radiation in large-scale real-world outdoor scenarios for real-time reconnaissance without prior knowledge of any sources or the terrain.

Measures to reduce the severe effects of radiation in such an event can be applied by rapid localization to allow the establishment of source contamination procedures. One possibility is that ground personnel enter the contaminated zone and use manual search techniques under strict protective measures. Unmanned vehicles (UGV, UAV) equipped with sensors can speed up the search and reduce radiation exposure. UAVs have the advantage of greater flexibility in terms of maneuverability, allowing larger areas to be covered when the terrain is uneven or difficult to access.

Radiation mapping methods project radiation intensities onto the ground to create a complete map of the area. This can be used to reveal areas of increased radiation and narrow the search area where the source is suspected. Explicit localization methods, on the contrary, address the problem of finding a point source directly [1,2]. Existing systems and methods utilize micro UAVs [3] and UAVs [4,5,6] at low altitudes <10 m or in combination with ground vehicles [7] to detect and localize point sources.

A high detection accuracy contributes to the fact that the source can be found more quickly and measures can be used in a more targeted manner. Depending on the flight altitude, accuracy of 17.8 m [8], down to less than 1 m at an equally low flight altitude of less than 1 m [5], is achieved, while obstacles which already have a strong influence on the result in the latter case.

Higher altitudes in contrary are crucial for being able to survey a larger area and quickly locate sources despite the presence of obstacles on the ground, but they pose other problems. It also increases the minimum distance to the source, which greatly reduces the measured intensity and allows background and terrestrial radiation to become dominant, making radioactive sources difficult to detect. Objects on the ground can additionally attenuate the radiation, resulting in unevenly distributed readings that further undermine mapping methods to locate sources.

For practical applications, a quick result is of great importance. However, data are often analyzed after the survey has been completed, e.g., in [5,7], which inevitably leads to delays. So far, only a few research results with real-time ability have been presented, e.g., in [3,4]. Nevertheless, concrete performance measures are still missing.

In this work, we therefore propose a real-time localization method to localize an a priori unknown number of radioactive sources in a search area of up to about 500 × 500 m with typical mission times of up to 20 min. In extensive experiments, we propose and evaluate our localization method in large-scale real outdoor scenarios with one Cs-137 and two Co-60 gamma-ray sources of category 3–4 according to the IAEA categorization of radioactive sources [9].

In [10] Maques et al., present state-of-the-art mobile radiation detection systems and list challenges for unmanned systems. We address 4 out of 9 identified items: (i) Improvement in data-to-decision-process is achieved as the proposed method is designed in such a way that no prior knowledge of radioactive sources is required. (ii) Real-time response is achieved as environmental sensors measurements are transmitted and processed on a standard PC immediately. (iii) Communication and bandwidth issues: Our method uses sub-sampled (LiDAR) data and experienced no bandwidth issues in a real environment. (iv) A reduced need for human supervision is achieved because our method is able to localize radioactive sources from altitudes up to 150 m, thus reducing collision risks with ground objects.

Specifically, the method is based on an optimization technique that takes into account the effective range of the gamma ray to derive source positions and strength, allowing simultaneous detection of multiple sources. The terrain topography is modeled continuously using a LiDAR scanner and attenuation effects are incorporated into the localization algorithm to compensate for terrain unevenness. We consider background radiation [11,12], thereby increasing the signal-to-noise ratio (SNR) at a given flight altitude and/or flight line spacing [11,13].

## 2. Related Work

There are three main system types of unmanned ionizing radiation detection and monitoring: Static ground-based, unmanned ground vehicles (UGVs), and unmanned aerial vehicles (UAVs). The first one requires installation of infrastructure assets ahead of time, with cost and space constraints limiting coverage area, detectable radiation intensities, and spatial resolution. In recent work based on stationary sensor networks, the idea of using multiple detectors to estimate source localization is considered. Research by [14] analyzes the performance of three different location-based detection approaches. The publication in [15] examined the trade-off between performance and cost factors of a distributed sensor network for the detection and localization of radiation sources based on Monte Carlo simulations in a theoretical analysis.

UGVs are more flexible, yet compared to UAVs, they are relatively slow and unfavorable terrain conditions might limit their operational scope. Especially in outdoor environments, this implies either frequent human intervention or elaborate schemes to identify traversable areas as, for example, demonstrated in [7]. Moreover, airborne systems for ground observation missions typically carry radiation detection equipment underneath, so that shielding effects by the vehicle itself can be eliminated. The use of mobile systems should be mentioned as an alternative. Flanagan et al. [16] propose the use of a mobile distributed sensor network using a public transportation system to protect large metropolitan areas from nuclear threats.

The main disadvantage of fixed or portable sensor networks is the use of a sometimes huge amount of detectors and the need to deploy the network before the attack or incident has even taken place.

Radiation monitoring consists of two distinct tasks, namely, mapping of the radiation distribution and localization of one or more point-like sources [8]. Both are important tools for competent authorities to implement countermeasures to RN incidents efficiently while minimizing exposure of citizens or emergency forces. Mapping usually determines the distribution of radiation on the ground, thus helping in identifying and securing the area of risk, for example, after the Fukushima Daiichi nuclear power plant accident in 2011 [17]. Typically, radiation intensity measured in-flight is adjusted for attenuation and distance in order to determine the corresponding radiation intensity at ground level [11,18]. Localization, on the other hand, also requires a (statistical) model to estimate source location(s) based on spatially scattered observations. It can be performed by exhaustively surveying the search area and estimating a posteriori likely source locations [4,5,7,19,20,21], or by applying greedy algorithms updating the current belief about source locations iteratively and adjust the measurement device’s trajectory accordingly [22,23]. The main advantage of greedy algorithms is their potential to significantly reduce mission duration. However, they also may be more vulnerable to premature termination in a multi-source setting. Another approach is to first generate a rough topographic mapping of the radiation field and then select regions of interest for detailed localization based on contour lines in the radiation map [8,24].

Some researchers incorporate terrain or environment features into their models. In airborne settings, this reduces errors introduced by varying distances between the ground and the aircraft/detector [11]. Additionally, shadowing of sources by structures or the terrain itself can be integrated into the localization model [1,25]. Terrain topography is usually obtained from geographic information system (GIS) products, or 3D scanning and photogrammetry/stereovision or LiDAR systems [1]. For instance, Gabrlik et al. [7] used an initial airborne photogrammetry scan to construct their own digital elevation model (DEM) and derive key mission parameters. In other cases, LiDAR [4,19] and stereovision [23] were used to assist source localization.

Some promising radioactive source localization algorithms have only been tested in silico [3,8,24], in small-scale laboratory environments [21,22,23], or in small-scale (<100 m^2^) outdoor environments [19]. It remains unclear whether these approaches can be successfully applied in larger, less controlled scenarios. Only few studies have been reported for UAV-based radioactive source localization in medium to large outdoor environments [4,5,7].

## 3. Materials and Methods

### 3.1. Radiation Physics and Dosimetry

In this section, we summarize the fundamental principles and physical laws that are essential to our radiation source localization approach.

#### 3.1.1. Dosimetry

Absorbed dose $D_{med}$ is the mean energy $dE_{abs}$ locally absorbed by an absorber material (med = medium) of density ρ during irradiation with ionising radiation divided by the mass *m* of the irradiated volume element $dm_{med}$. The SI unit of absorbed dose is given in joules per kilogram (1 J/kg = 1 Gy) and is also known as the Grey. In addition to absorbed dose, there is also exposure and kerma (kinetic energy of charged particles released in matter). The dose rates are formed as the differential quotient of the dose over time. A very good overview of these parameters is given in [26,27].

The equivalent dose *H* is of particular interest, to measure the stochastic risks of human radiation exposure. This is the product of soft tissue absorbed dose $D_{w}$ and quality factor *Q* at a point in the tissue. The quality factor is of dimension 1. This is determined by agreement for different radiation qualities in such a way that the same equivalent doses of different radiation qualities can be assessed in the same way under radiation protection aspects. For X-rays and gamma rays, Q=1 applies by definition.
(1)$$H=Q\cdot D_{w}.$$

Alternatively, the equivalent dose can be derived by measuring the total count rate $ZR_{k}$, which is measured in counts per seconds (CPS). This is the measurement we use in our radiation model. The count rate can be translated back to dose rate in [$\frac{Sv}{h}$] using a conversion factor α. It should be noted, however, that this conversion depends on many factors such as distance, source geometry and source composition, and many others.
(2)$$H=\alpha \cdot ZR_{k}.$$

In practice, this is used in airborne radiation mapping [1]. To determine the radioactive emissions from the ground, the cosmic background radiation and the radiation from the carrier platform itself must also be taken into account. The corrected total count rate is therefore:(3)$$ZR_{Corr}=SDI_{Meas}-SDI_{Back}-SDI_{Cosmic}$$
with $ZR_{Corr}$—the corrected total count rate, $SDI_{Meas}$—the measured count rate, $ZR_{Back}$—the background radiation of the carrier platform, and $SDI_{Cosmic}$—the cosmic background radiation, which is negligible at these altitudes.

Adjusting for the flight altitude leads to a significant improvement in the results, as already shown in [18]. This applies in particular to uneven or steep terrain, even if the measurements were carried out at a constant flight altitude. This correlation is formulated using the inverse square law and the attenuation.

In order to estimate the radiation on the ground from artificial sources, we also have to consider the terrestrial background radiation. In our work, we also take into account the attenuation by air, as well as the reduction in radioactive intensity proportional to the distance from which the measurement was made (inverse-square law) [27].

#### 3.1.2. Attenuation

In nuclear physics, attenuation refers to the decrease of ionizing radiation as it passes through a material [27]. According to the Lambert–Beer law, the intensity of a radiation decreases exponentially when it penetrates a material:(4)$$I_{x}=I_{0}\cdot e^{-\sigma \rho x}=I_{0}\cdot e^{-\mu x}.$$

Here, μ is the linear attenuation coefficient in [cm${}^{-1}$], σ is the mass attenuation coefficient in [cm${}^{2}$× g${}^{-}1$], ρ is the density of the material in [g × cm${}^{-}3$] and *x* is the thickness in [cm]. These attenuation coefficients basically depend on the material and on the radiation energy. However, the mass attenuation coefficient is very similar for many materials. The work in [28] shows a comprehensive mass attenuation coefficient database. Due to simpler handling when the material thickness is known, the attenuation is determined via the linear attenuation coefficient μ=σ·ρ. For this purpose, the material’s penetration length and the material-dependent linear attenuation coefficient must be determined. Since there are large differences in composition and physical properties, such as density, among natural as well as industrial materials, it is not possible to apply generally valid parameters; instead, approximate values are used. Examples of specific selected absorption coefficients have been given [29]. For airborne observations at low altitudes up to 150 m, the attenuation effect leads to a reduction of the radiation to about 1/2 to 1/4. Whereas, radiation reduction by the quadratic distance law has a considerably stronger effect. Only at altitudes close to 2000 m does attenuation become the dominant factor.

#### 3.1.3. Inverse-Square Law

The strength of a radioactive radiation rapidly decreases as the distance increases [27]. This is formulated in the inverse-square law, which states that the radiation is inversely proportional to the square of the distance from the point source. The prerequisite for this is that the spatial extent of the source is small compared to the distance. This law may therefore only be used to model point sources. The formula is:(5)$$I=\frac{I_{0}}{d^{2}}$$
where *I* is the radiation intensity at distance *d* from the source of intensity $I_{0}$.

### 3.2. Radiological Localization System

In this section, we describe the components of the real-time radiological localization system that was used in this work. We used a UAV as the carrier platform, where the sensors for radiological localization were mounted and fully integrated as a modular sensor bundle. The georeferenced data acquired by the system are transmitted to a ground station in real-time using a radio link. Figure 1 shows the carrier platform and the sensor bundle.

#### 3.2.1. The Carrier Platform

The RIEGL RiCOPTER-M [30] is used as the carrier platform (see Figure 1). Its sensor payload capacity is approximately 15 kg. Optional additional equipment such as a certified ADS-B transponder or a strobe light enable use in particularly critical environments (airports, restricted areas, night flights, etc.) The RiCOPTER-M has two short-circuit-proof power supplies for the sensor payload, which are powered by the main batteries of the aircraft. A 24 V supply is used for the VUX-1UAV laser scanner and a 12 V supply for the GAMON Drone gamma probe. Furthermore, an Ethernet interface is available for the laser scanner and another Ethernet interface for the gamma probe, which is used to retrieve the real-time data of both sensors. Additionally, an APX-20 GNSS/IMU system is mounted on the carrier platform. This system provides the necessary information for georeferencing the LiDAR and gamma data, in addition to providing the trajectory of the carrier platform. Synchronization (using GNSS timestamps) and georeferencing are performed on a Raspberry Pi 4 built into the RiCOPTER-M, which then transmits the data to the ground station via the RiCOPTER-M’s sensor data radio interface. An industrial switch connects all sensors via Ethernet on the RiCOPTER-M, which is also supplied with 12 V. The maximum flight time is 20 min with full sensor payload, the signal range is about 600 m visual line of sight.

#### 3.2.2. The Sensor Bundle

In this work, the sensor bundle concept was developed based on a modular approach. This way, it is possible to mount the sensor bundle not only on the carrier platform referred to in this work, but also, for instance, on a helicopter or any other airborne vehicle, provided that the relevant mechanical and electrical requirements are met. The focus of mechanical integration has mainly been placed on the RiCOPTER-M. The gamma probe was mounted on the mounting points of the VUX-1UAV and a small electronics box using an adapter plate, as shown in Figure 2. This setup prevents shading on the front exit glass of the VUX-1UAV LiDAR scanner. At the same time, this configuration has the advantage of ensuring a fixed mounting between the two sensors and the APX-20 GNSS/IMU system. To the left and right of the overall system structure, antennas were mounted which were used for data transmission. The overall weight of the sensor bundle amounts to 9.25 kg.

##### Positioning System

As positioning sensor, the high-performance GNSS-Inertial Solution with dual IMUs APX-20 GNSS/IMU [31] system was used. It consists of an electronic unit and the corresponding external IMU. This GNSS/IMU system weighs approx. 0.8 kg including housing and antenna. It is used to provide the flight trajectory as well as data for georeferencing the LiDAR and gamma data.

##### LiDAR Sensor

The LiDAR Sensor was used to generate a 3D point cloud of the mission area during the flight in real-time. In this work, we used the RIEGL VUX-1UAV [32] which was specially developed for use on UAVs and is characterised by its compact size and low weight (3.5 kg). It is a 2D TOF scanner with “rotating-mirror” principle with a field-of-view (FoV) of up to 360 degrees. With a fixed measuring rate of up to 550,000 measurements/second, rotation speeds of 10–200 Hz are possible. Up to 5 echoes per pulse can be detected. In this work, the LiDAR was set to operate on 200 kHz PRR (Pulse Repetition Rate) with a 100° Field of View.

##### Gamma Sensor

The purpose of gamma sensors is to measure radioactive radiation. Over distances, the radiation energy decreases considerably, which is why a very high sensitivity of the sensor system is of great importance for the detection of weak sources. In this work, we used the GAMON Drone (https://www.caen.it/products/gamon-drone/accessed on 30 August 2022) gamma probe which is a gamma radiation spectroscopy system based on scintillation detector and Geiger–Mueller counter.

This gamma probe sends status messages containing all cyclically collected measurement values, including the radiation measurement values Counts per Second (1 Hz), the dose rate integrated over several seconds (in the range of 10 s) and a spectrum (1 Hz). The spectrum consists of 2048 channels with a resolution of 1 keV, thus covering a range between 0 and 2 MeV. The gamma probe was configured to transmit status messages periodically (1 Hz) after start-up. In addition, the data were saved on the Raspberry Pi (locally on the UAV).

#### 3.2.3. Live Data Transmission

Data transmission from the carrier system to the ground station was conducted via radio link using UDP and TCP/IP. For the radio link between carrier platform and the ground station the SkyHopper PRO [33] was used. This is a Bi-Directional Data Link module for commercial/industrial drones with a 4.2 MHz bandwidth and 8 dBm transmitting power (Frequency: 2.315 MHz).

Generally, all data were recorded on the UAV. For LiDAR data, only a thinned measurement data stream was transmitted to the ground station using UDP. The degree of thinning can be established by selecting the scan line (line divider), point number (point divider) as well as the laser scanner pulse repetition rate. These settings are defined on the carrier system’s computer unit. In our work, we used the following parameters for reduction of the data rate:PRR: 100–200 kHz;Line Divider: 2;Point Divider: 3.

The density of the data set can be adapted in real time based on the settings mentioned above in order to use the full bandwidth of the data transmission.

Already georeferenced measurement or navigation data were transmitted to the ground station via a UDP/IP channel. To avoid frame fragmentation, transmission took place on the Ethernet network layer with a maximum packet size (Maximum Transmission Unit, MTU) of 1500 bytes.

### 3.3. Data Acquisition

#### 3.3.1. LiDAR Pointcloud

The laser scanner records 3D points of the scanned area with the coordinates, the exact time stamp and other properties. These data are used to create a 3D terrain model. In particular, these measurements, together with the UAV’s navigation data, allow the determination of the exact altitude above ground for each gamma measurement. The laser measurement data were transmitted in a proprietary transmission protocol in packets containing each about 60–70 laser points. The radio connection is designed for a max. data transmission of 6 Mbit/s. We thus achieved an effective transmission performance of approx. 25 kPts/s leaving some bandwidth for sensor control communication. The coordinates (x,y,z) are given in Cartesian coordinate system World Geodetic System 1984, WGS 84 (EPSG: 4978) with centimeter resolution. The coordinates were then transformed into the local metric coordinate system WGS 84/UTM zone 33 N (EPSG: 32633). The opening angle of the scanner was 100 degrees downward to prevent the creation of unnecessary points and to conserve bandwidth. Therefore, the width of the strip recorded on the ground during a flight corresponds to a little more than twice the flight altitude.

#### 3.3.2. Gamma Measurements

The gamma probe radioactivity readings as well as spectrum data were transmitted to the base station every second via the UAV radio module. In addition, the UAV GPS module was connected to the GAMON Drone sensor, enabling real-time location of the measured values. Each gamma measurement value thus received at the time of measurement the coordinates (x,y) in Cartesian world coordinates WGS 84 (EPSG: 4326) in degrees, the altitude *z* in meters, as well as the current time stamp defining the exact location and time of the measurement. The coordinates (x,y,z) were then transformed into the same metric coordinate system as the laser data (WGS84, EPSG: 32633).

#### 3.3.3. Radioactive Sources

In all of our test flights, we used up to three radioactive sources. These sources were provided by the CBRN Defense Center of the Austrian Armed Forces and were as follows:Two Co-60 sources;One Cs-137 source.

The cobalt sources (Co-60) differed slightly in strength with nominal acitvity of 489 MBq and 539 MBq (2.92 GBq and 2.65 GBq measured on 11.12.2008), and a much stronger cesium source (Cs-137) with nominal activity of 28.3 GBq (56.6 GBq measured on 1.8.1991). During the flights, the sources were used individually and also simultaneously. According to the “IAEA Safety Standards Categorization of Radioactive Sources for protecting people and the environment” [9], these radioactive source fall into the category 3–4. All sources could only be transported and laid out by trained personnel. Additionally, for safety reasons, all personnel had to maintain a safe distance at all times.

#### 3.3.4. Test Flights

In four days, we obtained 28 flights with different flight maneuvers, listed in Table 1. The flight patterns used in our tests are typically used to detect and locate sources on the ground:Calibration is an ascend and descend flight without any source to collect background radiation data at various altitudes;Meander covers an area with a rectangular grid. This technique is very effective for recognizing radioactive sources on a large area;Highest dose rate is a technique used to pin point a source by iterative crossing the source location;Lane search technique is used to detect sources along a path on ground, e.g., road. If an increased radiation is detected, the copter flies a loop around it;Cloverleaf pattern is similar to highest dose rate technique used to pinpoint a source.

### 3.4. Methodology

The goal of localization is to identify one or more point sources and to accurately determine their position. In some scenarios, it is possible to intuitively conclude the approximate position of a source on the basis of the measured radioactivity values, for example, if increased radiation and a typical spherical decrease in intensity of the radiation are already immediately recognizable in the recorded data. However, these preconditions by far do not apply to all scenarios. On the one hand, the background radiation varies strongly over the flight altitude, which is why increased radiation values can also result from lower flight altitudes. On the other hand, radiation can also be masked due to topology, so that reduced radiation values conversely do not exclude a source located in the immediate vicinity. In addition, the flight pattern itself also plays a role, e.g., the choice of line spacing. Finally, radioactivity levels from multiple sources may overlap, prohibiting an immediate source determination.

This requires the formulation of a mathematical model that describes radiation levels measured in the air as the result of radiation from radioactive sources located on the ground in order to make inferences about them. As with mapping, radiation decrease due to distance and attenuation effects must be included in the model.

In this work, we present a novel methodology allowing localizing one or multiple point sources. The method is based on an optimization process including modeling the background radiation as well as a radiation model considering the topology of the surveyed area.

#### 3.4.1. Background Radiation Model

In Section 3.1, we described how radiation can be measured. In this work, we assume that the measurements of the gamma sensor yields measurements as counts per seconds (CPS). It can be stated that the radiation registered by such a device measured in CPS, noted as *n*, is formed from the sum of the count rate from one or more sources $n_{S}$ and all other entries of the radiation background $n_{B}$. The latter are essentially always a present radiation exposure (natural background radiation). Therefore, we conclude that the total count rate *n* is given by:(6)$$n=n_{S}+n_{B}.$$

As noted in Section 3.1, background radiation can have several causes. Among the most important are:Natural terrestrial background radiation;Cosmic background radiation;Radiation from construction materials, e.g., the carrier platform.

The premise in our model is that we do not consider radiation resulting from cosmic background as it is only relevant for very high altitudes or altitude differences. Due to the altitude of the copter during the test flights of a maximum of 150 m, the cosmic background radiation can be neglected or considered as a constant component of the background radiation. In addition, we note that the construction material of the UAV does not contain any radiation material. The main contributing factor is therefore the altitude-dependent terrestrial background radiation.

In order to determine such altitude-dependent background radiation, we used an approach following the correlation described in Equation (Equation 3). To model the terrestrial background radiation, a calibration flight (before the actual measurement flight) was performed. Said flight had to be performed without any additional radiation source or outside the area of interference of a source. For this purpose, a climb and descent flight was flown to at least 100 m above ground or approximately to typical mission altitude. In order to minimize the effects of local ground-level natural variations in the background model, the ascent and descent should be as uniform as possible, so that measured values are distributed over all heights. We performed flights with a vertical speed of approx. 1 m/s.

From a practical point of view, a measurement flight almost always takes place at different heights above ground (even at constant flight altitude above mean sea level (AMSL)), depending on the terrain. The purpose of creating the background model is to determine the correct proportion of background radiation $n_{B}$ in the measured radiation for the respective flight altitude *h* to consequently compensate for this proportion.

Further on, we denote a set of measurements by X where $X_{i}={\left[x_{i},n_{i}\right]}^{T}\in X$ is the *i*-th measurement at position $x_{i}=\left(x_{i},y_{i},z_{i}\right)$ with a count rate given by $n_{i}$.

It is worth pointing out that, during the flight, we collected a point cloud which was then processed online in small chunks representing 10 s of data each. We applied the Simplified Morphological Filter [34] (using the default filter parameters suggested by the authors) to each chunk to extract ground points and construct a digital elevation model as the basis of our ground model $G_{z}\left(x,y\right)$. The ground model is used to determine the height above ground level $h_{i}$ for each measurement $X_{i}$:(7)$$h_{i}=z_{i}-G_{z}\left(x_{i},y_{i}\right).$$

According to (Equation 6), the remaining part $n_{i}-n_{B}$ is assumed to originate from artificial radiation source(s). To build the background model, the measurements of the calibration flight $X_{i}$ are considered as a function of height above ground (Figure 3). The background radiation decreases approximately exponentially with respect to the flight altitude *h*. Therefore, we can approximate the background radiation by using the exponential of a linear regression:(8)$${\bar{n}}_{B}\left(h\right)=10^{a+b*h}$$
with regression parameters $\left(a,b\right)\in R^{2}$. An example of data recorded by a calibration flight is depicted in Figure 3 where ${\bar{n}}_{B}\left(h\right)$ is visualized as a red solid line.

Additionally, the radioactive decay is a stochastic process which approximately follows a Poisson distribution [35]. Therefore, it describes the probability mass function of the measured count rates per second. Accordingly, the discrete probability $P_{\lambda }$ of measuring a certain count rate κ from a source with an average activity *I*, i.e., λ=I, is given by the following formula:(9)$$P_{\lambda }\left(\kappa \right)=\frac{\lambda^{\kappa }*e^{-\lambda }}{\kappa !}.$$

Hence, the background radiation can be described as a stochastic process, following the Poisson distribution, where as the parameter $\lambda ={\bar{n}}_{B}$, the mean background radiation determined for the particular flight altitude is used. This model shows high agreement with the actually measured values over all calibration flights. Furthermore, the percentiles for $B_{1}\left(h\right)$ and $B_{99}\left(h\right)$ are depicted in Figure 3 (red dashed lines).

The null effect (radiation that originates exclusively from the radiation background, i.e., natural radiation) of the measured radiation can be eliminated to obtain the radiation magnitude of additional sources:(10)$$R_{i}:=\max\left(n_{i}-{\bar{n}}_{B},0\right)\;\forall X_{i}\in X.$$

We denote as $R_{i}$ the positive residual radiation resulting by subtracting the mean background ${\bar{n}}_{B}$ from the measured radiation $n_{i}$. The residuals therefore determine the radiation foreground.

#### 3.4.2. Radiation Model

In order to infer the origin of the source and thus the radiation on the ground from the radioactivity measurements in the air, a radiation propagation model was developed. Given one or more specified sources, taking into account (Equation 6) and all the physical parameters discussed in Section 3.1 (background radiation, attenuation effect, inverse-square law, and stochastic distribution), this model allows to calculate the expected radiation at any point. In particular, it allows to describe radiation in the air at the actual measurement positions $x_{i}$. Here, the actual height above ground is already deduced from the laser measurements derived ground model $G_{z}$. Figure 4 shows a description of the air radiation calculation model.

We denote an artificial source with $S={\left[x_{s},n_{0}\right]}^{T}$ comprising of its position $x_{s}=\left(x_{s},y_{s},z_{s}\right)$ and its source strength $n_{0}$. In this model, we assume that each source is located on the surface. Thus, the altitude of the source is derived from the ground model $z_{s}=G_{z}\left(x_{s},y_{s}\right)$. The distance between any source and a target position is then given by the Euclidean distance:(11)$$d_{S,X_{i}}:=d\left(S,X_{i}\right)={∥x_{s}-x_{i}∥}_{2}.$$

According to (Equation 4) and (Equation 11), we can calculate the gamma radiation at any target point $X_{i}$ resulting from any source *S*:(12)$$n_{S}\left(S,X_{i}\right)=n_{0}\cdot \frac{1}{d_{S,X_{i}}^{2}}\cdot e^{-\mu d_{S,X_{i}}}.$$

This completes our radiation model according to (Equation 6), which gives an estimated count rate $\hat{n}$ at any target location $X_{i}$, assuming that one or more point sources $S_{j},j\in \left\{1...N\right\}$ of gamma radiation are applied:(13)$$\hat{n}\left(S_{N},X_{i}\right)=\sum_{j=1}^{N}n_{S}\left(S_{J},X_{i}\right)+{\bar{n}}_{B}\left(h_{i}\right),\;h_{i}=z_{i}-G_{z}\left(x_{i},y_{i}\right),\;\forall X_{i}\in X.$$

#### 3.4.3. Radioactive Source Localization

The starting point of the source localization is the correlation between the measured radiation $X_{i}$ and the radiation sources estimated by the radiation model described in Section 3.4.2. Generally speaking, each radiation source leads to a (statistical) increase of measured radiation at a certain position. Therefore, source localization aims at determining the position and the strength of each radiation source on the ground individually. In this respect, this method differs from mere mapping approaches in that one or more point sources are considered as the origin of increased measured radiation instead of an aerial radiation, e.g., due to contamination. For this purpose, we introduce the following localization optimization process.

##### Localization Optimization Process

We define an optimization process, by varying the model parameters (source position(s), source strength(s)), that finds a solution in which the deviation of the estimated radiation from the measured radioactivity is minimal across all measured values. Those parameters that give a minimum cost are consequently interpreted as a source. A two-step minimization problem is defined, whereas the first minimization step is to find a minimum cost at fixed source position $x_{s_{j}}$ by varying the source strength $n_{0}$ over all measurements $X_{i}\in X$. This problem can be written as a minimization problem of the following cost function:   
(14)$$\min_{n_{0}\in \left[n_{min},n_{max}\right]}f\left(S_{j},X_{i}\right),\;j\in \left\{1...N\right\},\;\forall X_{i}\in X,$$
whereas,
(15)$$f\left(S_{j},X_{i}\right)=err_{inlier}\left(S_{j},X_{i}\right)+err_{outlier}\left(S_{j},X_{i}\right).$$

To describe the individual terms of the cost function *f*, we first introduce the following set:(16)$$A:=\{\left(S_{j},X_{i}\right)|\hat{n}\left(S_{j},X_{i}\right)>B_{q}\left(h_{i}\right),\;j\in \left\{1...N\right\},\;\forall n_{0}\in \left[n_{min},n_{max}\right],\;\forall X_{i}\in X\}.$$

Therefore, A includes all target positions $X_{i}$ where the estimated radiation value $\hat{n}\left(S_{j},X_{i}\right)$ at given source strength $n_{0}$ exceeds the upper background percentile $B_{q}\left(h_{i}\right)$ of the background radiation model. It is further on named *inlier*. Consequently, all others $\bar{A}$ are *outlier*. We use a high percentile of q=0.99. Thus, an inlier refers to a target location where the source has an effect, i.e., it contributes to the gamma value. Vice versa, an outlier refers to a target position where the estimated gamma value does not exceed the background radiation. The cost function is defined as the sum of error from inliers $err_{inlier}$ and the error from outliers $err_{outlier}$ which serve as a regularization term to prevent over-fitting. Considering (Equation 16), the error from inliers is calculated as:(17)$$err_{inlier}=\sum_{(S_{j},X_{i})\in A}{\left|\hat{n}\left(S_{j},X_{i}\right)-n_{i}\right|}_{2}.$$

The error from outliers is calculated as:(18)$$err_{outlier}=\sum_{X_{i}\in \bar{A}}R_{i}$$
i.e., the sum of all positive residuals.

Note, that the distinction between inlier and outlier is used to establish a locality. Only inliers contribute to the error according to the error in gamma estimation, while outliers contribute with their residuals. This way, the optimization can be broken down to the search for the parameters of one source at a time.

Finally, the source strength $n_{0}\in \left[n_{min},n_{max}\right]$ defines the minimization problems parameter set. $n_{min}$ and $n_{max}$ are chosen as the minimum and maximum source strength for all positive residuals $R_{i}>0,\;\forall X_{i}\in X$. This can be calculated considering (Equation 6) and (Equation 10):(19)$$n_{0_{i}}=\underset{R_{i}}{\underbrace{\left(n_{i}-\bar{n_{B}}\right)}}\cdot d_{S_{j},X_{i}}^{2}\cdot e^{\mu d_{S_{j},X_{i}}}$$
with
(20)$$\begin{array}{cc} n_{min} & =\min_{X_{i}\in X}n_{0_{i}}, \end{array}$$
(21)$$\begin{array}{cc} n_{max} & =\max_{X_{i}\in X}n_{0_{i}}. \end{array}$$

In the second step of our minimization process, we use the source strength $n_{0}$, found through the first step (Equation 15), to vary the source location. For this, we introduce a discrete grid *G* that represents the target area for the localization process. An extensive search is applied by positioning artificial sources $S_{k,l}$ with source strength $n_{0}$ on the grid at positions $\left(x_{k},y_{l}\right),\forall \left(k,l\right)\in G)$, whereas the resolution determines the parameter space of the optimization process. Additionally, we assume that the source is located on the ground, i.e., $G_{z}\left(x_{k},y_{l}\right)$. In other words, we calculate step one (Equation 15) for each fixed position $\left(x_{k},y_{l}\right),\forall \left(k,l\right)\in G)$:(22)$$f_{k,l}:=f\left(S_{k,l},X_{i}\right),\;\forall \left(k,l\right)\in G,\;\forall X_{i}\in X.$$

Thus, we calculate a cost map defined on the grid *G*. Finding the minimum of the cost map finally provides us the location of an artificial source. In other words, the solution of our localization process. In order to find additional sources, local minima are also accepted as solutions.

For the evaluation of this model, this methodology was implemented by using the Python programming language. In Algorithm 1, the heuristic is depicted.
**Algorithm 1** Radiological source localization algorithm1:INPUT: Background model $\bar{n_{B}}$, Ground Model $G_{z}$, Measurements $X_{i}$2:**for **position=(k,l)∈G**do**3:    Compute $n_{min}$,$n_{max}$4:    Solve minimization problem $f_{min}=f\left(S_{k,l},X_{i}\right)$ with respect to $n_{0}\in \left[n_{min},step,n_{max}\right]$5:    Build Cost map $C_{k,l}=f_{min}$6:**end for**7:Find global and local minima of cost map *C* using peak search algorithm.8:OUTPUT: $\left(x_{i},y_{i}\right)$ locate radiation point sources

We note that from an applications perspective, the real-time capability of such a radiological localization system is essential. Algorithm 1 was implemented to be real-time capable on a state of the art processing unit. For our work, we used a Dell latitude 5531 comprising of the processor Intel Core i5-12600H with 2x 8192 MB DDR 4800MHz SDRAM. All tests confirmed that the system has real-time capability. Note that in this context, real-time capability is assumed when the processing time is less than the actual measurement flight.

## 4. Results

### 4.1. Test Site

The test flights were conducted at the military training site in upper Austria. This terrain includes different structures, elevation changes, buildings, roads, and vegetation that make it an ideal test site (Figure 5a). For regulatory reasons, the flight zone was set to a circle with a diameter of 900 m. All test flights were conducted within this zone.

To determine the source positions, the exact location was first measured with a GPS measuring device, and only then were the sources placed. Only those sources were laid out that were actually needed for the respective flight. Unused sources were stored safely so that they could not influence the measurement results. Figure 5b shows the locations of all sources. The exact coordinates are listed in Table 2.

### 4.2. Gamma Reconstruction

The background radiation was removed according to background model (Figure 3), thus revealing the radiation measured from any additional sources. Figure 6 shows the raw gamma measurements for a typical flight using meander pattern (a). Increased values at low altitudes, as they occur during take-off and landing of the copter due to the higher background radiation, shadow the effect of the source. Using Equation (Equation 10) already effectively eliminates background radiation (b). Note that the natural dispersion according to the Poisson distribution can still be observed.

### 4.3. Evaluation

To investigate the performance of the method, we evaluated the outcome by two metrics: First, we determined if the number of sources were detected correctly. Flights without any additional source (calibration flights) were evaluated as crosscheck. Consequently, for these flights, the outcome is expected to detect no additional source. It is also worth highlighting that any source far outside of the flying zone has been removed, if one should arise, i.e., local minima of cost map are accepted within the flying zone. Secondly, we determined the accuracy of the method by electing the Euclidean distance between the determined and the actual source position. Distances were calculated as the minimum distance to all sources and evaluated for detected sources individually.

Using this metric, all test flights were evaluated. Some test flights had the goal to determine the system limits and were therefore deliberately chosen to exceed the performance. Nevertheless, we chose to include them in the evaluation, as they give valuable insights of the system limitations. Results are shown in Table 3. The distance is denoted with “-” if there was no corresponding source and the method correctly detected no source. If a source was incorrectly not detected, this is indicated with “not detected”.

#### 4.3.1. Detection Performance

Test results show that the presence of any additional source was correctly detected for all flights. Correctly, no source was recognized for flights where no additional source was deployed. These are the calibration flights (no 1, 6, 7, 13, 23). Conversely, on all other flights where at least one source was present, at least one additional source was also discovered.

In five scenarios, the second source was also detected (flights 5, 12, 26, 27, 28). For flights 19–22, the second and for flight 22 also the third source was not detected. This is because we used three sources with very different power, namely, one Cs-137 and 1 to 2 Co-60 sources, which were also placed at a close distance (<100 m) from each other. Taking into account the radioactive decay of the nuclides since the sources were measured (Section 3.3.3), today, the nominal activity of the Cs-137 source is around a factor of 50 higher than that of the two Co-60 sources, so that the stronger Cs-137 source clearly outshined the other two Co-60 sources.

#### 4.3.2. Localization Accuracy

The first additional source was detected in all flights with distances from the true source position between 4.4 and 34.7 m (flight 5 with 72.1 m) with an average of 20.8 m. Flight no 5 significantly exceeded this average with distances of 64.4 m and 72.1 m. Here, the flight altitude was 118 m above ground, while the sources were 120 m away. Since at this flight altitude both sources are still not distinguishable from each other, the optimization resulted in a source position that lies between the two true sources. Consequently, the second source was also assumed to be at the wrong position. For all other flights, the true source position was determined, on average, to within 17.1 m.

Details of the optimization results are shown for some single-source flights and for dual-source flights in Figure 7, Figure 8 and Figure 9. In each row, the flight pattern is shown as an xz-projection (a) as well as an xy-projection to ground plane with the radioactivity readings (b) and the associated cost map (c). These flights cover an area from 0.3 to 26.9 ha. The position of the true source is plotted (red cross) in the cost map together with the optimized source position (red circle). The sources were found regardless of the flight pattern, but the selected flight pattern affects the area covered per unit time. Accordingly, flying at two different altitudes (flight 15) does not seem to provide a significant improvement in source detection over flying at only one altitude. Consequently, the flight time required for this can be saved or used to fly over larger areas.

The cost maps of flights 4 and 16 show a slightly skewed cost distribution with lower values in the direction of the takeoff and landing point. Both flights also have lower gamma values compared to the other flights (2, 15). A possible explanation for this is that the source radiation propagation for the gamma values of the climb and descent at an acute angle to the ground plane is attenuated by additional factors (vegetation, buildings) and consequently overestimated in the calculation. To verify this, we removed the climb and descent of the copter from the data for comparison for both flights and used only the measured values at the target altitude (760 m) and recalculated the cost map and source positions. This view corresponds to a mission scenario in which the copter is launched well outside the effective range of a source, as we would expect for real missions. That way, the source position was determined much more accurately (Figure 8).

The following examples show results for two and three sources (Figure 9). For flight 12 with low height above ground (approx. 60 m) and large source distances of 182 m, both sources were well detected. The second and third source in scenario 22 were not detected, as described earlier, due to the strong differences in intensity with simultaneous small distances between the sources.

### 4.4. Limitations

#### 4.4.1. Detection Limit

The detection limit of a source is determined by the distance to the detector. The relevant cut-off point is where the radiation intensity of the source becomes indistinguishable from the background radiation. Our radiation model defines this level as the 99th percentile of the background radiation. Figure 10a shows radiation levels as a function of flight altitude (vertical climb directly above a source) for background radiation and sources of various intensities with concrete measurements of Co-60 and Cs-137 sources as overlay. The mean background radiation (solid black line) indicates the lower boundary of measurable radiation intensity. The upper percentile within which 99% of the background radiation is expected to be found is indicated as a dashed black line. The measured values of two vertical climb flights are plotted as dots. The Co-60 source (orange dots) can be detected up to a flight altitude of about 130 m, while the much stronger Cs-137 source (blue dots) can still be detected at flight heights (distances to source) of far more than 300 m.

#### 4.4.2. Separation of Sources

In principle, the system is multi-source-capable, with no limitation to the maximum number of sources that can be detected simultaneously. As our algorithm identifies local optima for source locations in the radiation field, we investigated the minimum distance required between two sources to separate them successfully. The used criterion mandates a drop of at least 10% in measured radiation intensity exactly between the sources at constant flight altitude. In other words, the sources remain distinguishable as long as a superimposition of their respective partial radiation fields still reveals two clear intensity peaks in the overall radiation field. With increasing flight altitude, however, the required distance between sources increases. Figure 10b illustrates this relationship. It shows the parameter space within which two equally strong sources can be distinguished (below the orange and blue lines). Background radiation obviously blurs the radiation field and thus impairs the source resolution. In essence, both sources must also be clearly recognizable against the background radiation, which is why stronger sources can be recognized as individual sources at even higher altitudes.

## 5. Discussion

In this work, we have demonstrated the applicability of our method to locating multiple radioactive sources in real-world use cases in relatively large areas of about 500 × 500 m which is a significant improvement over earlier works in terms of flight altitude and the size of the observed area [3,4,5,8,19,21] and real-world validation [22,23,24].

In our approach, the UAV travels along a predefined search pattern at an altitude of up to 150 m. Source detections are reported within the convex hull of the UAV’s trajectory, which is why the UAV has to pass the sources on two different trajectories. This can be easily achieved by standard flight patterns such as meanders or Archimedes’s spiral, provided that enough measurements are collected within the effective range of the source. We control this factor by suitable selection of flight strip spacing. Consider that localization will not work in the case of a simple fly-by along a straight line because symmetries in sensor and environment configurations lead to ambiguous results. Automatic path finding algorithm or adaptive routing [3,8,22,23] could be combined with our real-time method to further improve detection time.

The proposed method works independently of the number of sources and the terrain model, but relies on knowledge of the terrestrial background radiation. If a calibration flight cannot be performed, e.g., due to recent contamination, one may approximate the model using previously recorded data. Over- or underestimation of the residual gamma values may occur if the expected background radiation is not fully representative of the target area. Overestimation is expected to have little effect on the estimated position of detected sources, given that the method looks for local optima, which in this case do not undergo any change. Underestimation, on the other hand, leads to a corresponding loss of sensitivity.

The target area could be expanded significantly, but its maximum extent is mainly governed by two factors, namely, the maximum mission time of the UAV/aircraft, and processing power. While our method is not dependent on the type of aircraft selected, grid-based source localization scales poorly for larger areas [1] and thus creates a potential bottleneck. As a counter measure, one could decrease the grid resolution proportionally at the expense of localization accuracy. If high localization accuracy is still required, the rough estimates of source locations obtained from the wide area search could be used to plan subsequent localization missions in smaller target areas. We consider the time penalty incurred by the subsequent small-scale localization mission as relatively small compared to the duration of wide area search. Additionally, the lower-quality results from the wide area search will still be available in real-time, which is important for practical application. Note that we did not implement any such routines, nor did we conduct a detailed performance analysis because we were able to obtain satisfactory results on a standard office notebook within a short time of about one to two minutes. Such analysis, however, could guide mission parameter selection in the future.

Source detection and localization performance are largely governed by the strength of the source itself, since with increasing distance between source and detector, the SNR of source and background radiation decreases to the point where a source becomes indistinguishable from background noise. This limits both flight altitude and the distance between flight lines. Similarly, reducing flight altitude improves the ability to detect weak sources and discriminate multiple radiation sources located close to each other.

Our algorithm searches for the smallest possible number of sources to describe the measured radiation field, taking into account the range of the sources based on the SNR. This allows to find any number of sources. However, due to the mutual superimposition, sources that are very close to each other can be perceived as a single stronger source and thus falsify the results of the source localization. This situation could always be resolved by lowering flight altitude to an appropriate level.

We compute the distance from the UAV to the ground based on actual terrain measurements recorded in-flight utilizing a LiDAR sensor. This is an obvious improvement over simpler models where the terrain is assumed to be a horizontal plane [3,5,8,17,24]. Alternatively, this particular improvement could also be achieved by using readily available GIS products, even though the available resolution and thus accuracy of the terrain model will typically be somewhat lower. Our method requires real-time terrain mapping only when historical data are unavailable or outdated due to large geological mass movements.

The presented implementation of our radiation source localization method ignores attenuation by terrain features such as hills and vegetation, buildings, and other man-made structures (i.e., all objects are assumed to have the same attenuation properties as air), since we consider the source to be unobscured by ground objects from above. However, in order to assign different attenuation coefficients to different volumes in space, classes should be assigned to the 3D terrain model. A range of classification algorithms have been presented throughout the literature, with notable examples ranging from geometric or morphological analysis [36,37,38] to more sophisticated deep learning approaches [39,40,41]. Terrain classification could be especially useful in scenarios with highly variable terrain or if sources are located close to massive structures that are at least partially blocking direct line of sight during flyovers. We hypothesize, though, that even a rudimentary classification of terrain and terrain features into e.g., ground, buildings/structures, and vegetation could already considerably improve localization performance in such challenging setups and we plan to investigate such approaches in the future.

## 6. Conclusions

In this work, we present a radiation source localization method suitable for real-world (airborne) missions in relatively large target areas of more than 200,000 m^2^. A UAV system is presented that locates multiple radioactive gamma sources with IAEA classification category 3–4 from typical UAV operating altitudes of up to 150 m with an average distance of 17.1 m. The average computing time in all experiments was about 2 min, from the time of measurement to the graphical presentation of the results. Our method iteratively solves an optimization problem to fit radiation sources into the observed radiation field and is not bound by a maximum number of detectable radiation sources, nor is any *a priori* knowledge about the number of sources required. The algorithm can be run on a standard office notebook, thus making it suitable for real-time applications and time critical missions in both civil and military RN incident response. Integrated LiDAR scanning allows considering terrain altitude to improve localization accuracy, and it also opens up new opportunities to determine different attenuation coefficients for individual volumes in space.

## 7. Patents

The findings presented in this work are part of two patents registered on the 31 August 2022 at the German Patent and Trade Mark Office (DPMA). Official file number: 10 2022 121 951.8 and 10 2022 121 953.4.

## Figures and Tables

**Figure 1 sensors-22-09198-f001:**
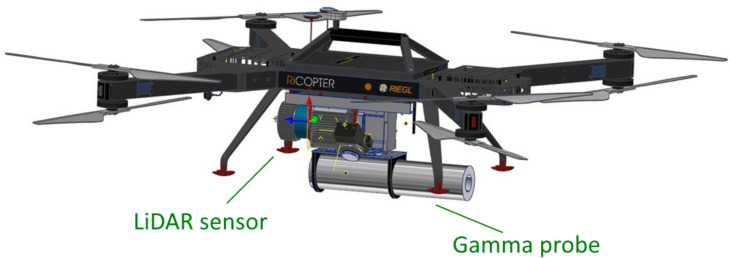
Carrier platform and sensor bundle comprising LiDAR sensor and gamma probe.

**Figure 2 sensors-22-09198-f002:**
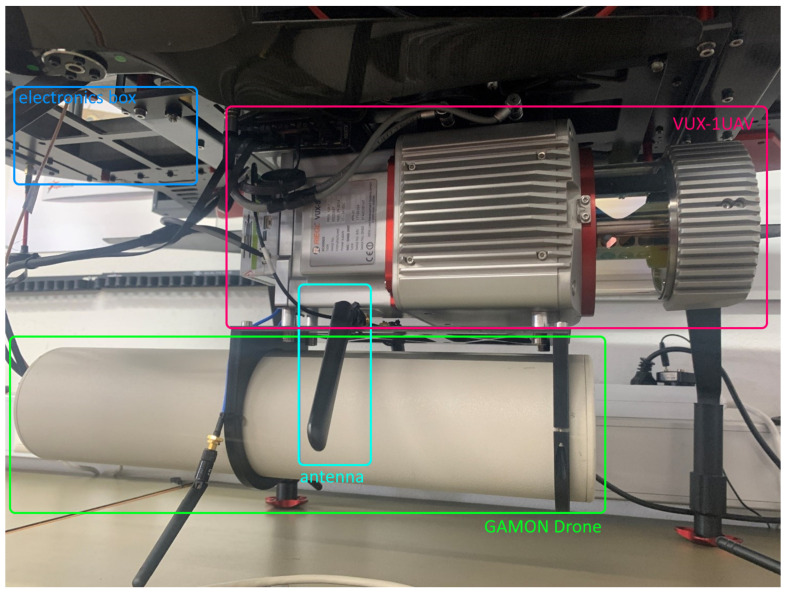
LiDAR sensor RIEGL VUX-1UAV (top, red) and CAEN Sys GAMON Drone gamma radiation spectroscopy system (green), electronics box (blue), antenna (cyan).

**Figure 3 sensors-22-09198-f003:**
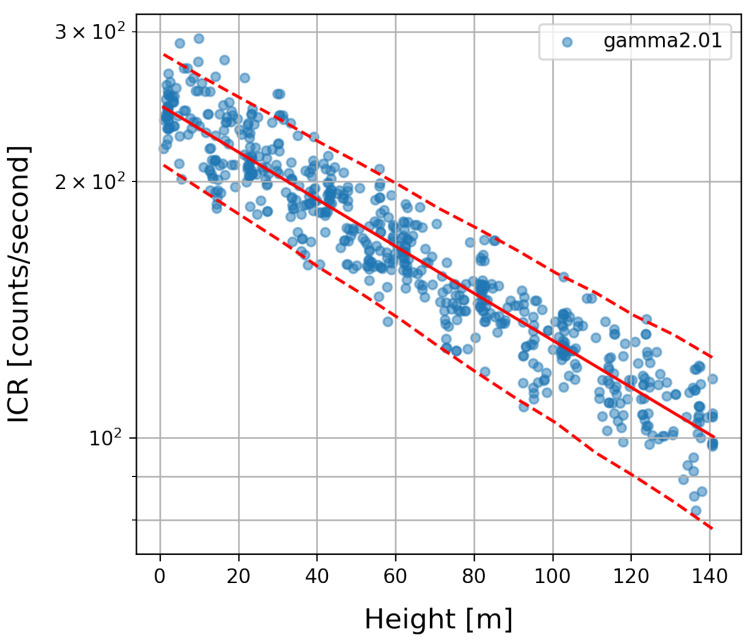
Radiation background modeling. The measured counts per second are plotted at their corresponding measuring heights.

**Figure 4 sensors-22-09198-f004:**
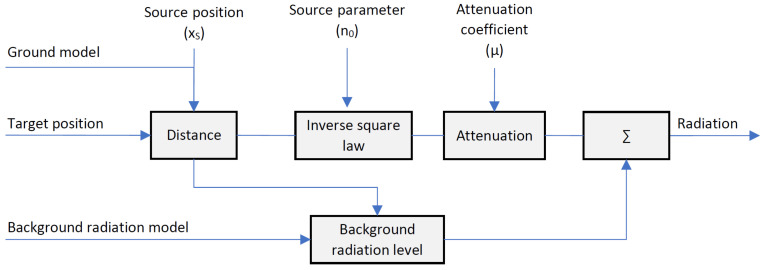
Radiation model: calculation of the expected radiation at any point.

**Figure 5 sensors-22-09198-f005:**
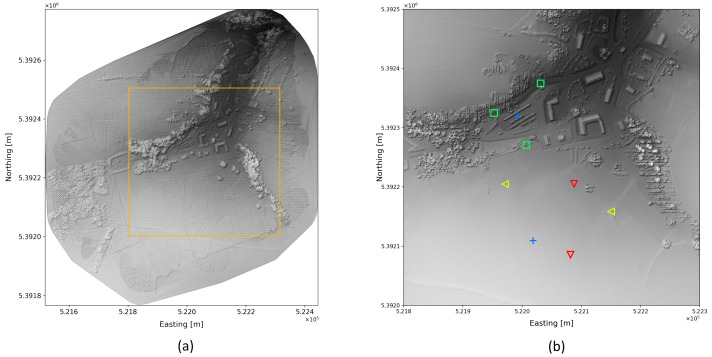
Projection of LiDAR points of the test area (**a**). Position of the radioactive sources during test flights (**b**) for flights 1–5 (red triangle), flights 6–12 (yellow triangle), flights 13–22 (green squares), and flights 23–28 (blue cross).

**Figure 6 sensors-22-09198-f006:**
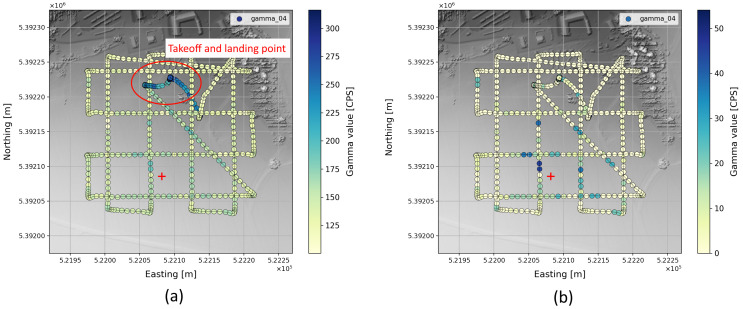
Application of background model. Original measurement data along the trajectory (**a**) naturally exhibit Poisson distributed scattering analogous to background radiation. After removal of the background radiation, only the additional radiation from radioactive sources remains (**b**). The true source position is marked with a red “+”.

**Figure 7 sensors-22-09198-f007:**
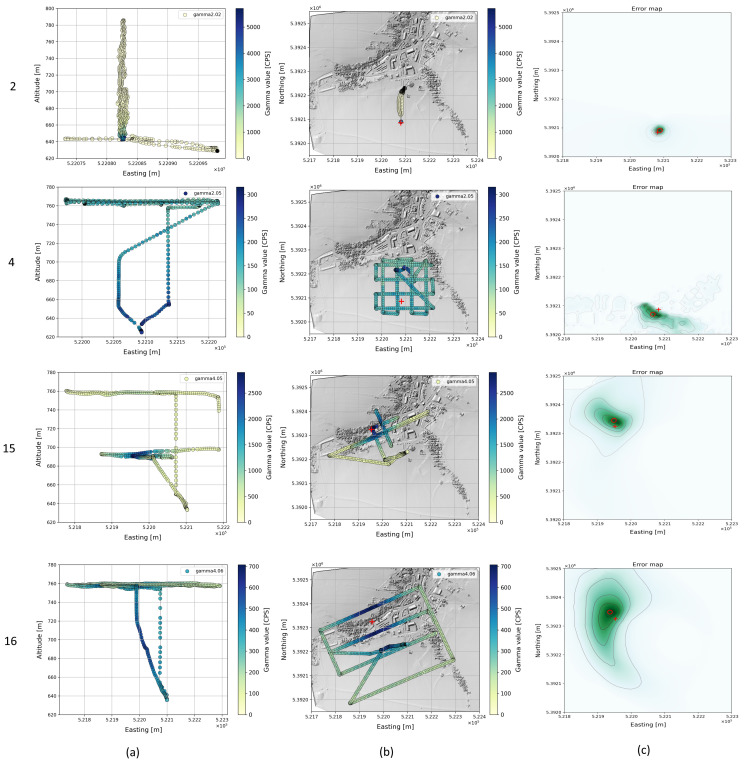
Flight pattern for selected flight (2, 4, 15, 16) with single radioactive source projected to xz-plane (**a**) and projected to ground xy-plane (**b**). The gamma measurements are color-coded. The corresponding cost map is shown in (**c**).

**Figure 8 sensors-22-09198-f008:**
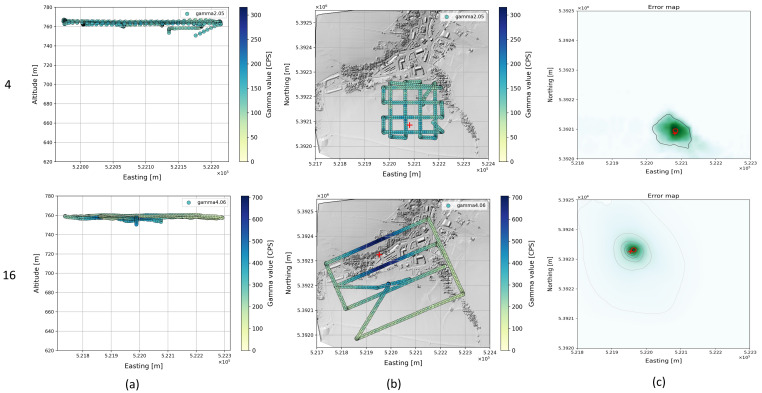
Flight pattern without climb and descent for selected flights (4, 16) with single radioactive source projected to xz-plane (**a**) and projected to ground xy-plane (**b**). The gamma measurements are color-coded. The corresponding cost map is shown in (**c**).

**Figure 9 sensors-22-09198-f009:**
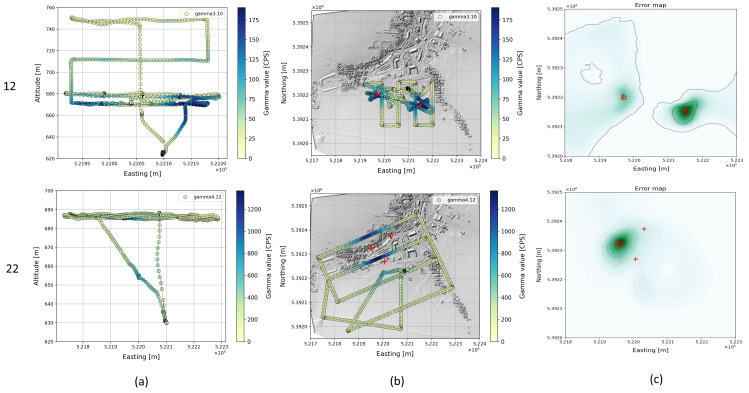
Flight pattern for selected flights (12, 22) with multiple radioactive sources projected to xz-plane (**a**) and projected to ground xy-plane (**b**). The gamma measurements are color-coded. The corresponding cost map is shown in (**c**).

**Figure 10 sensors-22-09198-f010:**
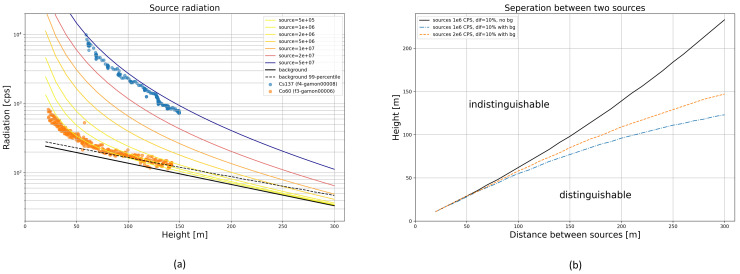
Detection limit by flight altitude with measurements from two flights up to 150 m above ground (**a**) and source separation limits (**b**).

**Table 1 sensors-22-09198-t001:** Overview of the test flights.

No	Source(s)	Maneuver	Duration [s]
1	-	Calibration	823
2	Co-60	Calibration	573
3	Co-60	Meander	680
4	Co-60	Meander	539
5	Co-60, Co-60	Meander	516
6	-	Calibration	792
7	-	Calibration	321
8	Co-60	Meander	649
9	Co-60	Highest dose rate	508
10	Co-60	Highest dose rate	563
11	Co-60	Cloverleaf pattern	486
12	Co-60, Co-60	Cloverleaf pattern	756
13	-	Calibration	336
14	Cs-137	Lane search technique	489
15	Cs-137	Lane search technique	475
16	Cs-137	Lane search technique	659
17	Cs-137	Meander	699
18	Cs-137	Cloverleaf pattern	714
19	Cs-137, Co-60	Lane search technique	556
20	Cs-137, Co-60	Lane search technique	658
21	Cs-137, Co-60	Cloverleaf pattern	585
22	Cs-137, Co-60, Co-60	Lane search technique	685
23	-	Calibration	2939
24	Co-60	Lane search technique	865
25	Co-60	Meander	538
26	Co-60, Co-60	Meander	699
27	Co-60, Co-60	Lane search technique	421
28	Co-60, Co-60	Highest dose rate	475

**Table 2 sensors-22-09198-t002:** Source positions.

Used in Flight(s) no.	Source	Latitude	Longitude	Distance to First Source [m]
2–5	Co-60	48.68250	15.30008	-
5	Co-60	48.68142	15.29999	120
8–12	Co-60	48.68210	15.30090	-
12	Co-60	48.68250	15.29850	182
14–22	Cs-137	48.68358	15.29826	-
19–22	Co-60	48.68403	15.29932	93
22	Co-60	48.68309	15.29898	76
24–28	Co-60	48.68353	15.29879	-
26–28	Co-60	48.68164	15.29914	212

**Table 3 sensors-22-09198-t003:** Localization results.

No	Max Flight Height [m]	Number of Sources	Distance 1st Source [m]	Distance 2nd Source [m]	Distance 3rd Source [m]
1	153	0	-	-	-
2	153	1	8.3	-	-
3	141	1	27.7	-	-
4	148	1	20	-	-
5	118	2	64.4	72.1	-
6	133	0	-	-	-
7	109	0	-	-	-
8	135	1	34.7	-	-
9	98	1	14.4	-	-
10	91	1	10.4	-	-
11	99	1	23.7	-	-
12	124	2	4.4	3.4	-
13	134	0	-	-	-
14	143	1	15.7	-	-
15	146	1	15.1	-	-
16	150	1	18.2	-	-
17	144	1	33.3	-	-
18	144	1	21.6	-	-
19	112	2	15.1	not detected	-
20	115	2	7.9	not detected	-
21	110	2	16	not detected	-
22	77	3	17.6	not detected	not detected
23	130	0	-	-	-
24	108	1	31.1	-	-
25	97	1	8.6	-	-
26	79	2	10.8	33.4	-
27	71	2	13.5	20.2	-
28	65	2	9.1	11.3	-

## Data Availability

The data presented in this study are available on request from the corresponding author. The data are not publicly available because they were recorded in a research project. Data presented in this work specific to the environment and topography (i.e.: LiDAR sensor data) is not avialable due to restrictions of the research project. For a minimal data set in context of the environment and topography, we refer to basemap (accessed on 30 August 2022).

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
