# Peer review of "Real-Time Gamma Radioactive Source Localization by Data Fusion of 3D-LiDAR Terrain Scan and Radiation Data from Semi-Autonomous UAV Flights"

_sensors, 2022, doi:10.3390/s22239198_

Round 1

Reviewer 1 Report

This article is interesting, well-structured, and of great application value, and presents a very far-reaching study. My comments are as follows:

There are editing errors on lines 86 and 11, and these sentences are missing a period at the end.

Line 132 "where" should capitalize "Where".

Line 326 "(7" is missing ")".

Is there an editorial error in line 352 "taking into account 7"?

In the introduction I would like to know more about the advantages of this article compared to previous studies.

Reviewer 2 Report

1. There is not a related work section in the paper. It’s better to separated introduction and related work into two different sections.

2. Please give a brief introduction to the respective symbol of Equation 3.

3. The introduction of Figure 1 does not make the reader understand the location and distribution of each sensor. You can draw the lead and write the sensor name in the figure. In Figure 5b, appropriate amplification can be done to match the size of Figure 5a.

4. To prevent radiation, a drone is used, say how long the Carrier Platform can last and how far the signal it can receive.

5. Note formatting issues such as if head titles need to be bold. In section 2.2.3, the correct writing of the unit should be the dBm. In the first paragraph of page 10, note the case issues.

6. In the positioning results of Table 3, the position can be adjusted and arranged according to certain rules, so that it is more convenient for readers to compare between the data.

Reviewer 3 Report

In the article I did not find any scientific and practical novelty, as the goals initially set incorrectly, that is, it is too apart from reality. Given where the authors work, I have the impression of a primitive bloated work at the level of a technical report.

Relevance does not match with the chosen methods: in the relevance the authors talk about radiation incidents (when radionuclides fall out in large spots and the source is extended), and the methods and model chosen for a point source, which can introduce a fairly large uncertainty in the measurement results.

Introduction is rather misleading regarding the tasks to be solved.

Why in the section of materials and methods you describe the values not used further in the article, it is necessary to reduce this section to the required number of used values. You should delete the sections Dosimetry, Attenuation, Inverse-square law, except for the selected model. In general, the model has long been known in the field of dosimetry and protection from ionizing radiation.

The way the authors describe the inverse-square law depresses me, because it means they are bad experts.

The inverse-square law looks like H ~ 1/r^2, not as you have H ~ exp(coef*r). Please remove this mistake.

The description of methods and models is a rewriting of basic knowledge from textbooks.

Some references are incorrectly cited (essentially missing).

The reasoning on Lines 300-307 does not match what is written in relevance. The authors write that just their study is applicable to cases of radiation accidents in densely populated areas, where the main source is not terrestrial radiation but radiation from building materials, which varies widely around the world. The fact that the authors do not take into account cosmic radiation, this has long been studied and calculated by other researchers - there is nothing new here.

Lines 319-322 describe the standard procedure for calibrating the detector and terrain-specific determination, in the case where simulations are not used.

The authors themselves have not written in the conclusion that they have developed anything new, both scientifically and practically.

To summarize, the problem solved in the work concerns the search for very highly active point sources. To solve such problems a method based on the law of inverse squares has long been developed. The authors used it in their study, however, this method allows only to localize a highly active gamma source, which does not require such complex computational procedures and mathematical constructions. The authors have exaggerated the description of the search procedure from 2-3 pages to more than 20, which is of course a big disadvantage, and the mathematical constructions to describe simple things make the material difficult to understand for readers.

The solution uses known knowledge from nuclear physics, dosimetry and ionizing radiation protection, and the basic basics of statistical data processing. Nothing in terms of developing new sensors or measurement methods has been done by the authors.

Unfortunately, in my opinion, there is nothing meaningful here for Sensors journal. I recommend the authors to try to significantly reduce the article in terms of technical description (data transmission protocols, technical characteristics) and send the manuscript to the MDPI Mathematics journal.

P.S. Nevertheless, the shortcomings that I found, which definitely need improvement can be found in the attached PDF file by opening it in a browser (Firefox/Chrome).

Round 2

Reviewer 3 Report

The authors slightly improved the article, but only in terms of removing the description of obvious things not used further in the paper (terms from the dosimetry discipline). I did not notice any other improvements in the article. I still consider the work to be at the level of a routine engineering task. The problem of searching for lost gamma-active sources has been known and solved since the last century, is included in all textbooks of dosimetry and protection from ionizing radiation, and is solved by students in their respective specializations.

If the authors had solved the task, that they stated at the beginning of the introduction: «This need is made stronger by the increasing threat of RN attacks on soft targets and critical infrastructure in densely populated areas» then it would be a worthwhile article, solving a complex engineering problem - searching for gamma sources on the terrain with different building densities and types (residential, industrial, military, etc.).

The authors claim that they have invented a new methodology to search for sources at unmanned aircraft altitude, but in fact they only perform mapping with additional processing of the obtained data using already known methods (gamma background subtraction, surface smoothing and solving the minimization problem of a known function).

I stand by my opinion that the article should be rejected, and I recommend the authors to seriously refine the article by answering the following important questions:

1) How to perform a calibration flight in a densely populated area if it is already contaminated?

2) The disadvantages are the same as static monitoring stations require calibration flights in advance?

3) What happens in case of rain/snow (different weather conditions) during the calibration/measurement flight?

4) What if your sample of terrain (not radioactive, for calibration) may not be representative (e.g. different building density)?

5) How can you account for the buildings themselves as radioactive objects, as well as for differences in building materials and ground composition of the terrain?
